# Intrinsic Motivation: Knowledge, Achievement, and Experimentation in Sports Science Students—Relations with Emotional Intelligence

**DOI:** 10.3390/bs13070589

**Published:** 2023-07-14

**Authors:** Isabel Mercader-Rubio, Nieves Gutiérrez Ángel, Sofia Silva, Guilherme Furtado, Sónia Brito-Costa

**Affiliations:** 1Department of Psychology, Faculty of Education Sciences, Universidad de Almería, La Cañada, 04120 Almería, Spain; imercade@ual.es; 2Research Group in Social and Human Sciences (NICSH), Coimbra Education School, Polytechnic Institute of Coimbra, 3045-043 Coimbra, Portugal; sofiace@esec.pt; 3Centro de Estudos Interdisciplinares (CEIS 20), Universidade de Coimbra, 3000-457 Coimbra, Portugal; 4Centro de Investigação em Educação de Adultos e Intervenção Comunitária (CEAD), 8005-139 Faro, Portugal; 5Applied Research Institute, Polytechnic Institute of Coimbra, 3045-043 Coimbra, Portugal; guilhermefurtado@ipc.pt; 6Research Unit for Sport and Physical Activity (CIDAF), Faculty of Sport Sciences and Physical Education (FCDEF-UC), 3040-248 Coimbra, Portugal; 7Human Potential Development Center (CDPH), Polytechnic Institute of Coimbra, 3045-043 Coimbra, Portugal

**Keywords:** motivation, emotional intelligence, athletes, regulation, students

## Abstract

In this paper, we have focused on the Self-Determination Theory, paying special attention to intrinsic motivation, which is understood as the motivation that leads the subject to perform a task without the expectation of obtaining an external reward. In the field of sport, motivation is one of the most studied variables and one of the most researched, since it is closely related to the reasons that lead the athletes to start, maintain, and abandon sports practice. The main objective of this study is to analyze the relationship between intrinsic motivation (IM) and emotional intelligence from the theoretical contributions of the Self-Determination Theory. The specific objectives are to analyze attention, clarity, and emotional regulation, as well as intrinsic motivation to acquire knowledge. Additionally, we aim to explore the relationship between attention, clarity, and emotional regulation and the intrinsic motivation to achieve something. Lastly, we investigate the correspondence between attention, clarity, and emotional regulation and the intrinsic motivation to experience stimuli. The sample consists of 163 undergraduate and master’s students related to Physical Activity and Sports Sciences, studying at a Spanish public university, located in the southeast region of Andalusia—specifically, in the city of Almería. The participants had a mean age of 20.33 years. In terms of gender, 70.9% (*n* = 117) were men and 27.9% (*n* = 46) women. In terms of degree, 76.7% (*n* = 147) were undergraduate students and 23.3% (*n* = 18) were postgraduate students. The Sports Motivation Scale (SMS/EMD) was used to assess intrinsic motivation, and the TMMS-24 was used to assess emotional intelligence. The main findings of this research demonstrate the existence of a relationship between the three dimensions of emotional intelligence (emotional attention, emotional clarity, and emotional regulation) and intrinsic motivation (intrinsic motivation to know something, intrinsic motivation to achieve something, and intrinsic motivation to experience stimulation). These findings emphasize the importance of emotional intelligence for intrinsic motivation.

## 1. Introduction

When a person practices sport, their primary purpose is the paradigm or improvement of bodily, physical, and psychological situations, without forgetting issues such as the personal interrelationships that are established through the practice of sport and the achievement of objectives and results [1]. Sport psychology has been consolidated in recent years as a science that investigate various variables of a psychological nature that are related to the athlete’s performance, with motivation being one of the most studied [2,3,4,5,6,7,8,9]. Therefore, when we talk about motivation, we refer to a variable that has been postulated as one of the main topics addressed within sport psychology [10], since it is assumed as a basic aspect for acquiring responsibility and adherence to sport practice, and therefore we can establish it as the determining variable at a psychological level [11]. In this sense, motivation is understood as the tendency that leads a person to perform an action with a specific intention [12,13].

Currently, there are many theories related to motivation, but the most popular and widely accepted is the Self-Determination Theory (SDT). This theory offers a theoretical model that corresponds to an explanatory theory of human motivation. It aims to provide answers about the human behaviors that individuals perform volitionally or in a self-determined manner, specifically examining the degree to which people engage in actions with a high level of reflection and a sense of choice [14]. This theory has provided a significant contribution to the understanding of motivation and has been applied in different fields, including sports. Its main contributions in the realm of sports involve determining the level of athletes’ involvement based on different psychological aspects that regulate behavior [14]. Within this framework, motivation is understood as a multidimensional psychological construct that fluctuates across different levels of intensity, responds to various purposes and intentions, and can be influenced by different factors [14].

In addition, the contributions of the TAD identify three levels of motivation: amotivation, extrinsic motivation, and intrinsic motivation. At the extreme end of the motivation spectrum, we find demotivation, which occurs at both the intrinsic and extrinsic levels. It corresponds to the lowest level of self-determination and is characterized by a complete lack of intention to act [14].

An example in sports psychology is when athletes face situations in which they do not feel capable or safe. Athletes at this level participate in sports without having affective, social, or material goals, and they experience negative emotions such as apathy, depression, or a sense of incompetence [15,16].

On the other hand, extrinsic motivation is driven by the rewards obtained when engaging in a behavior. It is characterized by the external satisfaction that comes from prizes or rewards [14]. A clear example of this type of motivation in the field of sports is when individuals play sports for social recognition or due to internal or external pressures.

Lastly, intrinsic motivation is driven by the pleasure derived from engaging in an activity, accompanied by feelings of competence and self-realization [14]. Within intrinsic motivation, three subtypes are identified. The first is intrinsic motivation towards knowledge, where athletes engage in an activity for the pleasure and satisfaction they experience while trying to learn. The second subtype is intrinsic motivation towards mastery, where athletes participate in an activity for the pleasure of improving and surpassing themselves. The third subtype is intrinsic motivation towards stimulation, where athletes engage in an activity to experience sensations associated with their senses [17,18,19].

Due to the numerous studies in the field of non-competitive physical activity that emphasize the importance of developing strategies for intrinsic motivation [9,10], our research has focused on the Self-Determination Theory, with particular attention to intrinsic motivation. Intrinsic motivation is defined as the drive that leads individuals to engage in a task without expecting external rewards [14]. The primary motivation for the individuals is centered around learning, skill development, skill practice, or the inherent pleasure derived from the activity itself [15,16]. Furthermore, various types of intrinsic motivation have been identified, constituting a triadic contribution comprised of the following [17,18,19]:

Intrinsic motivation to acquire knowledge: This pertains to engaging in activities that inherently bring pleasure or satisfaction through learning and exploration.

Intrinsic motivation to achieve goals: This corresponds to the satisfaction derived from creating or perfecting an action.

Intrinsic motivation for sensory stimulation: This relates to activities that stimulate individuals, primarily in the domains of art, aesthetics, amusement, or enjoyment.

In this way, within the field of sport and from the contributions of sport psychology, there are various investigations based on the contributions of the Theory of Self-Determination [20,21,22,23,24,25,26], which have had the purpose of studying the motives that lead the behaviors of the sportsperson to start, maintain, and abandon sport [27,28,29,30].

Another of the psychological constructs that has been widely studied in sport psychology is emotional intelligence [31,32,33,34,35,36,37,38,39,40,41,42]. Regarding the conceptualization of this construct, we find two major approaches that explain its general contributions: on the one hand, we find the mixed models, and on the other hand, the ability models. In this research our focus has been on the ability models [43].

From the ability models, Emotional Intelligence is based on mental abilities, with a standardized character [44] which serve a dual purpose: firstly, the regulation of one’s own emotions through cognitive processes [45], excluding aspects related to personality from this definition. In other words, based on these models, emotional intelligence is conceived as a set of abilities to reflect on emotions, and their potential to influence and guide thinking [43]. These abilities can be acquired, improved, learned, and trained. Currently, the Emotional Intelligence model is widely accepted, disseminated, and researched—particularly in countries like Spain, where it is considered the model with the strongest empirical support and scientific rigor [46,47].

These models are composed of three dimensions:Emotional attention: comprehended as the ability to collate and evaluate one’s own and others’ feelings.Emotional understanding: understood as the ability to recognize, classify, and investigate one’s own and others’ feelings.Emotional regulation: understood as the ability to assimilate, defer, and reflect on one’s own and others’ feelings.

Regarding the benefits and contributions of sports psychology, there are numerous investigations that have taken emotional intelligence as a central theme, the results of which indicate that higher levels of emotional intelligence are directly and positively correlated with variables such as motivation [48,49,50,51], life satisfaction [52], identified regulation, introjected regulation and external regulation [49], self-efficacy [53,54], basic psychological needs [55,56], interpersonal relationships [57], low levels of anxiety [58], or with psychological skills [59] and subjective well-being [60,61,62,63,64].

However, focusing specifically on the relationship between emotional intelligence and intrinsic motivation, several studies have been conducted to gain a deeper understanding of the influence of emotional intelligence on motivation [35,36,37,38]. The study of these issues is considered highly important, which further justifies the significance of this research.

Firstly, intrinsic motivation is associated with desirable attitudes and values in sports practice, as well as better learning outcomes [65,66]. It also demonstrates a direct correlation with greater task persistence and overall well-being, spanning from childhood through adolescence [67,68].

Secondly, within sports activities and practices, intrinsic motivation is linked to a positive attitude towards sport [69], greater involvement [70,71], active participation [72], high levels of physical activity [73,74], positive social and interpersonal experiences [75], increased effort [76], and high levels of enjoyment [77,78]. Thus, intrinsic motivation shows a positive relationship with emotional attention, empathy, and emotional regulation [79,80,81].

For this reason, emotional intelligence is of great interest in the sports field, and the skills related to emotional intelligence (such as self-knowledge, emotional self-regulation, self-motivation, social skills, or empathy, etc.) serve as tools athletes need to successfully navigate various challenging situations. In other words, emotional intelligence equips athletes with the necessary control to manage their own emotions effectively.

Similarly, within the realm of university education, which is the context of our sample, research has demonstrated that intrinsic motivation is closely linked to the ability to comprehend and acquire knowledge about personal and interpersonal emotions. It also encompasses the capacity to experience novel or unconventional emotions and effectively express one’s own emotional states [49,50,51,52,53,54,55,56,57,58,59,60,61,62,63,64,65,66,67,68,69,70,71,72,73,74,75,76,77,78,79,80,81,82,83].

Therefore, it is the very essence of the ability models to conceptualize emotional intelligence as a set of skills that can be acquired, improved, learned, and trained. This perspective has guided our decision to adopt these models for our research.

Furthermore, previous studies [84] have indicated that intrinsic motivation has a statistically significant impact on each of the dimensions that constitute emotional intelligence, namely attention, clarity, and emotional regulation. Based on these findings, the hypotheses of our research have been formulated to address the following research question: Can the implementation of mental and emotional training aimed at enhancing the development of emotional intelligence among future professionals in physical activity and sport sciences enhance students’ intrinsic motivation?

This was the rationale behind selecting the sample in order to examine the state of our students and future professionals in the field of physical activity and sport sciences with respect to these matters. We conducted an analysis of their self-perceptions regarding both intrinsic motivation and emotional intelligence. These findings also serve as feedback to identify the aspects that our students are knowledgeable about and proficient in, as well as those areas where further training is needed in sports psychology. The lack of research addressing the intersection of intrinsic motivation and emotional intelligence limits our understanding of the psychological processes involved in sport and how emotional intelligence influences these dynamics. By filling this gap, we aim to build a better knowledge base, enabling the development of more informed and effective approaches and strategies to improve and motivation of athletes.

## 2. Materials and Methods

### 2.1. Method

Taking these theoretical contributions as a reference, the main objective of this study is to analyze the relationship between intrinsic motivation (IM) and emotional intelligence based on the theoretical contributions of the Self-Determination Theory. The specific objectives include the analysis of attention, clarity, and emotional regulation, as well as the intrinsic motivation to acquire knowledge. Additionally, the research investigates the correspondence between attention, clarity, and emotional regulation and the intrinsic motivation to experience stimuli.

To this end, we set out the following hypotheses:

**Hypothesis** **1:**
*The three dimensions of emotional intelligence (attention, clarity, and emotional regulation) exhibit a direct and positive relationship with intrinsic motivation to know something.*


**Hypothesis** **2:**
*The three dimensions of emotional intelligence (attention, clarity, and emotional regulation) exhibit a direct and positive relationship with intrinsic motivation to achieve something.*


**Hypothesis** **3:**
*The three dimensions of emotional intelligence (attention, clarity, and emotional regulation) exhibit a direct and positive relationship with intrinsic motivation to experience stimuli.*


This research is framed within a quantitative, exploratory, descriptive, and explanatory method, utilizing two validated and standardized questionnaires as a data collection technique [85,86,87].

The first questionnaire used is the Trait Meta Mood Scale (TMMS 24) [88], which is a self-report measure assessing self-perceived emotional intelligence through 24 items. The scale is theoretically based on ability models [44] and consists of eight items that evaluate emotional attention, emotional clarity, and emotional regulation. Sample statements include, “I pay close attention to my feelings,” “I am clear about my feelings,” and “I have a lot of energy when I feel happy”.

The second instrument used is the Sports Motivation Scale (SMS/EMD) [8,19], which measures motivation across through seven factors: FACTOR 1 is related to the identified regulation; FACTOR 2 is related to introjected regulation; FACTOR 3 is related to external regulation; FACTOR 4 is related to demotivation; FACTOR 5 is related to intrinsic motivation to experience stimulation; FACTOR 6 is related to intrinsic achievement motivation; FACTOR 7 is related to the intrinsic motivation towards knowledge.

This scale consists of a total of 28 items, providing a comprehensive evaluation of motivation beyond intrinsic and extrinsic categories. Sample questions include, “I participate in sport for the enjoyment that it gives me,” “I participate in sports for the prestige of being an athlete,” and “I participate in sports because I often wonder about it since I am not achieving my goals.

This study corresponds to an ex post facto design, which fulfills the requirement of including students pursuing degrees related to physical activity and sport sciences. This type of study and design aligns with a methodological approach widely accepted and utilized in educational and psychological research to explore the level of agreement and perception among participants regarding a specific topic [89]. The sample was selected using convenience sampling, considering the students who attended class on the agreed day, as communicated with the teacher in advance.

### 2.2. Participants

The sample consisted of 163 students from different degrees related to physical activity and sports sciences (both the degree in physical activity and sports sciences and the master’s degree in research in physical activity and sports sciences) who study at a Spanish public university, located in the southeast of the Spanish country, specifically in Andalusia, in the city of Almería. The mean age was 20.33 years, with a standard deviation of SD = 3.44. Regarding gender, 70.9% (*n* = 117) were men and 27.9% (*n* = 46) women. The main reason for choosing a sample made up of students was to know what the starting situation is, and the type of knowledge that future professionals possess in their initial training. In this sense, we must highlight that within the Spanish university system, the studies leading to the teaching of physical education are divided into three large areas: the first refers to the official degree title in physical education and physical education sciences, the second refers to the specific mention in physical education within the studies of the official degree title in primary education, and the third refers to postgraduate or master’s studies, where there is both the official master’s degree in physical activity and sports sciences, as well as the master’s degree in teacher training with a mention in physical education. In this case, this research takes as a sample the three large areas of university training in terms of the teaching of physical education, with a distribution of the sample as follows: 76.7% (*n* = 147) were studying undergraduate, either the official degree in Physics, Activity, and Sports Sciences, or the official title of Primary Education with a mention in Physical Education, while 23.3% (*n* = 18) were enrolled in related postgraduate studies; in this case, the official master’s degree in physical activity and sport sciences, and the master’s degree in teacher training with a mention in physical education. The research was carried out during the 2021/2022 school year.

Table 1 shows the main socio-demographic data of the sample in terms of sex, year, degree, and age. All participants are university students who are active in sport. None of them are an elite athlete. Among the socio-demographic items, we asked the sample how many hours a week they spent practicing sport. None of the participants stated that they did not play sport. Overall, 36.9% (N = 60) said they practiced sport between 1 and 2 h a week, 40.9% (N = 67) said they practiced sport between 3 and 5 h a week, and 22.2% (N = 36) said that they practiced sport more than 5 h per week.

### 2.3. Ethics Statement

This study was approved by the Institutional Review Board of the University of Almeria (UALBIO2022/035). For its development, the participants were firstly informed about the main objective of this research and the total confidentiality of the data was guaranteed. Prior to the completion of the questionnaire, all participants signed the official informed consent of the University of Almeria (Spain).

### 2.4. Instruments

The instruments used in this work correspond to the following:We used the Trait Meta Mood Scale (TMMS 24) [88]. It is a validated and standardized instrument which through a scale that measures self-perceived emotional intelligence in the form of a self-report consisting of 24 items. Theoretically, it is based on the ability models of emotional intelligence, through the measurement of the three dimensions of emotional intelligence—attention to emotions (items 1 to 8), emotional clarity (items 9 to 16), and emotional regulation (items 17 to 24) [90]—based on a Likert-type scale (1 to 5: where 1 means not at all agree and 5 means strongly agree).The psychometric properties are adequate, and for this research a Cronbach’s alpha = 0.84 was obtained for the overall scale. Adequate results were also obtained for each branch of emotional intelligence (perception, α = 0.90; clarity, α = 0.90; regulation α = 0.86) and adequate test-retest reliability: perception = 0.60, understanding = 0.70, and regulation = 0.83 [44]. Specifically, in this research, we obtained an overall Cronbach’s alpha = 0.84, and for each of the dimensions the following scores: perception, α = 0.84; clarity, α = 0.80; regulation α = 0.84.Sports Motivation Scale (SMS/EMD) [8,19] (see Appendix A). This corresponds to a validated and standardized instrument which through a scale measures motivation through seven factors. Regarding each one of them, FACTOR 1 is related to the identified regulation; FACTOR 2 is related to introjected regulation; FACTOR 3 is related to external regulation; FACTOR 4 is related to demotivation; FACTOR 5 is related to the intrinsic motivation to experience stimulation; FACTOR 6 is related to intrinsic achievement motivation; FACTOR 7 is related to the intrinsic motivation towards knowledge. We must indicate that in this work we randomly chose the scores referring to factors 5, 6, and 7. In addition, the answers are issued through a Likert-type scale (1–7: where 1 means not at all agree and 7 strongly agree).

We obtained a Cronbach’s alpha = 0.73. Moreover, it shows a high reliability (Cronbach’s alpha) for each subscale (intrinsic motivation to experience stimulation, α = 0.79; intrinsic motivation to get things, α = 0.79; intrinsic motivation towards knowledge, α = 0.79).

### 2.5. Procedure

The research procedure is summarized in the following phases:

PHASE 1: Permission to conduct the research was sought from the Institutional Review Board of the University of Almeria, which gave us their consent.

PHASE 2: Through email we arranged with a lecturer from each course a day and time to attend class and provide the questionnaires. This was completed during the first or last 10 min of class. In the email sent to the teachers, they were also informed of the purpose of this research.

PHASE 3: In each of the classes we attended, the questionnaires were filled in by all the students who, after explanation of the purpose and guarantee of the confidentiality of the data, decided to participate.

STAGE 4: All participants signed the official informed consent of the University of Almeria (Spain) after being informed of the data protection protocol.

STAGE 5: The total sample size is conclusive due to the total number of students who gave prior informed consent and decided to participate in our research. The N of the study universe refers to the cluster from which the information is drawn. In its totality, it corresponds to 277 participants. To perform this calculation, the total student enrollment in each of the four years of the official degree in physical activity and sport sciences, the physical education track within the official degree in primary education, the master’s degree in physical activity and sport sciences, as well as the specialization in physical education within the official master’s degree in teacher training, were considered as reference points.

### 2.6. Data Analysis

Initially, the data analysis was descriptive (mean, standard deviation, and bivariate correlations). The purpose of this type of statistic is that it allows us the exploratory and descriptive analysis of the data [91]. In this case, the different tests were interpreted from an analytical perspective through the statistical indicators established for it.

After that, reliability analysis and structural equation modeling (SEM) were performed to differentiate the concrete relationships in the hypothetical model. The reasons for using SEM correspond to the purpose of evaluating both multiple and crossed dependency relationships [92], as well as to identify which are those factors that explain the relationships between the different variables, assuming that from these models, there are no variables more relevant than others [93], to finally test the hypotheses [93,94,95,96,97,98,99,100].

Specifically, a non-recursive structural model was applied. It was decided that it should be non-recursive because while in recursive models the explanation is ordered asymmetrically in one direction, while in non-recursive models, relationships appear that invert the order of causality, establishing reciprocal relationships. Therefore, non-recursive structural models are more realistic than recursive ones [101].

After that, both the variances between the previous factors were calculated. As can be seen in Table 2, before the model was validated, the values of different parameters were determined, as well as the goodness or fit indices. To support or question the proposed model, we consider different fit indices and their adequacy: TLI above 0.95 (Tucker–Lewis index), SRMR (standardized root mean square residual), and RMSEA below 0.08 (root squared error). half). means of approach) [91]. All these statistical tests were performed with SPSS version 26 and the statistical analysis programmer R. Finally, the model was accepted, and the hypotheses were accepted, concluding the same.

To test each of these hypotheses, the following steps were followed:

**Hypothesis** **4:**
*The three dimensions of emotional intelligence (attention, clarity, and emotional regulation) present a direct and positive relationship with the intrinsic motivation to know something.*


To test this hypothesis, the variables involved were the following:emotional care: items 1 to 8 of the TMMS24 questionnaire. Creation of all items in a new variable named AE.emotional clarity: items 9 to 16 of the TMMS24 questionnaire. Creation of all items in a new variable named CE.emotional regulation: items 17 to 24 of the TMMS24 questionnaire. Creation of all items in a new variable called RE.intrinsic motivation to know something: items 2–3–23 and 27 of the SMS/EMD scale, called FACTOR 6. The methodology and the tests to verify it were as follows: first, the mean score for each factor was calculated. Next, the correlations between each of the dimensions of emotional intelligence and the intrinsic motivation to know something were calculated, and finally, the non-recursive model was carried out.

**Hypothesis** **5:**
*The three dimensions of emotional intelligence (attention, clarity, and emotional regulation) present a direct and positive relationship with the intrinsic motivation to achieve something.*


To test this hypothesis, the variables involved were the following:emotional care: items 1 to 8 of the TMMS24 questionnaire. Creation of all items in a new variable named AE.emotional clarity: items 9 to 16 of the TMMS24 questionnaire. Creation of all items in a new variable named CE.emotional regulation: items 17 to 24 of the TMMS24 questionnaire. Creation of all items in a new variable called RE.intrinsic motivation to achieve something: items 8–12–15–20 of the SMS/EMD scale, called FACTOR 7. The methodology and the tests to verify it were as follows: first, the mean score for each factor was calculated. Next, the correlations between each of the dimensions of emotional intelligence and the intrinsic motivation to know something were calculated, and finally, the non-recursive model was carried out.

**Hypothesis** **6:**
*The three dimensions of emotional intelligence (attention, clarity, and emotional regulation) present a direct and positive relationship with the intrinsic motivation to experience stimuli.*


To test this hypothesis, the variables involved were the following:emotional care: items 1 to 8 of the TMMS24 questionnaire. Creation of all items in a new variable named AE.emotional clarity: items 9 to 16 of the TMMS24 questionnaire. Creation of all items in a new variable called CE.emotional regulation: items 17 to 24 of the TMMS24 questionnaire. Creation of all items in a new variable named RE.intrinsic motivation to experience stimuli: items 1–13–18–25 of the SMS/EMD scale, called FACTOR 5. The methodology and the tests to verify it were as follows: first, the mean score for each factor was calculated. Next, the correlations between each of the dimensions of emotional intelligence and the intrinsic motivation to know something were calculated, and finally, the non-recursive model was carried out.

## 3. Results

The links between each of the dimensions of emotional intelligence (AE/CE/RE), intrinsic motivation to know something (F5), intrinsic motivation to achieve something (F6), and intrinsic motivation to experience stimulation (F7), are presented in Table 2, where the correlations between these variables are shown, reflecting the reciprocity between them. The bivariate correlations were calculated based on the scores obtained. The closer the correlation coefficient is to +1, the stronger the association between the variables. A correlation coefficient of +1 indicates a perfect positive linear relationship, suggesting a direct association between the variables. In this particular case, the positive correlations observed between the variables reflect their reciprocity, indicating that they are associated in a direct sense.

### Structural Equation Model

The hypothetical model, represented by Figure 1, Figure 2 and Figure 3, exhibited satisfactory global fit indices, indicating a good overall fit of the model. The significance level (*p*-value) was less than 0.001, indicating a statistically significant fit.

The Root Mean Square Error of Approximation (RMSEA) was 0.05, suggesting a reasonable fit of the model to the data. The Goodness of Fit Index (GFI) was 0.965, indicating a high level of agreement between the model and the observed data. Furthermore, the incremental or comparative fit indices provided additional support for the proposed model. The Comparative Fit Index (CFI) was 0.941, indicating a good fit when compared to a model of independence or no relationship between the variables.

Similarly, the Incremental Fit Index (IFI) was 0.945, further confirming the adequacy of the proposed model. Overall, these fit indices suggest that the hypothetical model demonstrates a good fit to the data and provides valuable insights into the predictive relationships among the variables.

The three non-recursive models presented highlight the relationships established in the structural equation model specified below:(a)Intrinsic motivation to know something (F5) was positively correlated with emotional attention (=0.04, *p* < 0.001), emotional clarity (=0.04, *p* < 0.001), and emotional regulation (=0.24, *p* < 0.001).(b)Intrinsic motivation to know something (F6) was positively correlated with emotional attention (=0.06, *p* < 0.001), emotional clarity (=0.04, *p* < 0.001), and emotional regulation (=0.68, *p* < 0.001).(c)Intrinsic motivation to experience stimulation (F7) was positively correlated with emotional attention (=0.06, *p* < 0.001), emotional clarity (=0.01, *p* < 0.001), and emotional regulation (=0.92, *p* < 0.001). Therefore, based on these results, we can affirm that emotional intelligence is closely related to all types of intrinsic motivation.

These results highlight the fact that those athletes with higher levels of emotional intelligence also show greater pleasure or satisfaction in learning and exploring, in creating or perfecting an action, and are more comfortable in actions mainly related to art, aesthetics, fun or enjoyment.

As can be observed from the scores obtained, the three dimensions of emotional intelligence exhibit a direct and positive relationship with the three types of intrinsic motivation: the intrinsic motivation to know something (F5), the intrinsic motivation to know something (F6), and intrinsic motivation to experience stimulation (F7). These findings are consistent with previous research [92,94] that has also highlighted the reciprocal and existing connection between emotional intelligence and intrinsic motivation. Intrinsic motivation refers to the driving force derived from the pleasure and satisfaction experienced while engaging in an activity, accompanied by feelings of competence and self-actualization [14].

Based on these findings, it can be affirmed that emotional intelligence is closely related to intrinsic motivation. The results indicate that athletes with higher levels of emotional intelligence also experience greater pleasure and satisfaction when engaging in activities such as learning, exploring, creating, or perfecting actions. Furthermore, they feel more at ease when participating in endeavors associated with art, aesthetics, fun, or enjoyment. It can be concluded that athletes who possess higher levels of self-determination also tend to exhibit elevated emotional intelligence [27,28,29,30,31,32], thus demonstrating the link between self-determination theory and emotional intelligence [79,80,81]. Consequently, we can describe this athlete profile as having a positive attitude towards sports practice [69], displaying high levels of involvement [70,71], being active participants [72], engaging in significant levels of physical activity [73,74], fostering positive relationships with others [75], exerting greater effort [76], and deriving considerable enjoyment from their pursuits [77,78].

## 4. Discussion

The purpose of this research was to examine the contributions of Self-Determination Theory, and specifically intrinsic motivation to know something, intrinsic motivation to achieve something, and intrinsic motivation to experience stimulation and their relationship with emotional attention, emotional clarity, and emotional regulation. From the three hypotheses proposed, we can affirm that all of them are fulfilled, since our results show that those athletes with higher levels of emotional intelligence also present greater pleasure or satisfaction when learning and exploring, when creating or perfecting an action, and feel more comfortable in actions mainly related to art, aesthetics, fun, or enjoyment. However, we cannot fail to emphasize that our sample has a very limited number of participants, which does not allow us to generalize these results.

Furthermore, the main theoretical contribution of our research lies in the fact that the results provided are congruent with the results provided by previous research, which highlighted that emotional intelligence directly and positively predicts greater intrinsic motivation [92,94]. In other words, they demonstrate the coexistence of a precise relationship between different types of intrinsic motivation and emotional intelligence. As has been indicated by other recent research, we can draw a logical inference from the obtained results, pointing to a clear and positive correlation between levels of emotional intelligence and levels of self-determination [27,28,29,30,31,32].

However, from an educational point of view, our results go further, and going back to the research question posed, the following question is proposed: Could the use of mental and emotional training to promote the development of emotional intelligence of future professionals in physical activity and sport sciences increase students’ intrinsic motivation? The answer is affirmative, and these are the main findings of our research, since intrinsic motivation and emotional intelligence can be considered as factors that favor adherence to sporting activities among university students studying degrees related to physical activity and sport sciences [95].

In turn, the main contribution of this study is the contribution to knowledge about emotional intelligence in the field of sport in general, and from the initial training of sports science students. We must consider it as a practical way of action for the proposal and creation of different strategies to work on emotional intelligence and motivation in students of degrees related to physical activity and sport sciences. These strategies encompass not only cognitive aspects but also psychological and emotional factors, aligning with the athlete’s competencies within the specific context in which they operate [96]. Through the strategic management of pivotal psychological variables in this field, this approach facilitates comprehensive decision-making in the implementation of various actions that effectively produce desired outcomes [97].

## 5. Limitations

It is essential to exercise caution when interpreting the results of this research, primarily due to the limited sample size, which does not allow for generalization of the results. However, in this regard, we would like to point out the fact that the N of the research universe, understood as the total conglomerate from which the information is extracted, was a total of 277 participants. To do this, we add the total number of students enrolled in each of the four courses of the official degree in physical activity and sports sciences, the total number of students enrolled in the mention of physical education of the official title of primary education, the total number of students enrolled in the master’s degree in physical activity and sport sciences, and the total number of students enrolled in the physical education specialty within the official master’s degree in teacher training.

In addition, we use the margin of error calculator, and specify the total population size, the desired confidence level (the standard value used by most researchers is 95%), and the sample size. The score obtained was 3.20%. In this sense, the further the percentage is from 50%, the smaller the margin of error turns out to be. For this reason, although the size of the sample is limited and we cannot generalize the results in the general population, they do correspond to representative results at the local level in which the study was carried out.

Therefore, it would be ideal to conduct future studies with larger sample sizes to verify these observations and address this current limitation. Future research directions will also focus on investigating potential discrepancies based on the type of sport practiced. Moreover, additional studies will aim to explore variations in relation to the degree of professionalization within each sport among future participants. Furthermore, it is imperative to conduct longitudinal studies to investigate the effect of the passage of time, without forgetting the study of the influence of variable differences between sex and between different ages.

Additionally, expanding the scope of participant groups to include practicing coaches and teachers will contribute to a more comprehensive analysis. It is crucial to consider whether the research encompasses athletes training in individual or group sports. Moreover, incorporating information-gathering techniques such as interviews and focus groups will offer valuable insights. Lastly, it is essential to acknowledge that this research is limited in its cross-sectional design, warranting caution when generalizing the findings.

## 6. Conclusions

In summary, the present research provides empirical evidence supporting the proposed model’s hypothesized relationships between the three dimensions of emotional intelligence (emotional attention, emotional clarity, and emotional regulation) and intrinsic motivation (including intrinsic motivation to acquire knowledge, intrinsic motivation to achieve goals, and intrinsic motivation to seek stimulation). These findings strengthen our understanding of the intricate interconnections between emotional intelligence and intrinsic motivation, demonstrating the relationship between the ability to collate and evaluate one’s own and others’ feelings, the ability to recognize, classify, and investigate one’s own and others’ feelings, and the ability to assimilate, defer, and reflect on one’s own and others’ feelings. In addition, the findings suggest that the feeling of greater pleasure or satisfaction when learning and exploring, when creating or perfecting an action, and feeling more comfortable in actions mainly related to art, aesthetics, fun, or enjoyment in university students belongs to degrees related to the sciences of physical activity and sport. In conclusion, emotional intelligence is deemed significantly important for intrinsic motivation. Based on these findings, we strongly emphasize the need to prioritize emotional training, in addition to physical and cognitive training, in the education of future professionals in physical activity and sport sciences right from the beginning. This comprehensive approach will empower students to effectively navigate various sporting and work-related situations. Furthermore, it underscores the paramount role of sports psychologists in supporting individuals, highlighting the importance of their presence in optimizing emotional well-being and performance in the field of sports.

Furthermore, our results reveal the relationship between the three dimensions of emotional intelligence and intrinsic motivation among university students in the field of physical activity and sport sciences. This demonstrates that the treatment of emotional intelligence and intrinsic motivation are important aspects to develop and cultivate in future teachers and coaches, to help them understand and support the intrinsic motivation of their students or athletes. This, in turn could contribute to increasing the commitment, satisfaction, and performance of students and athletes in physical and sports activities. Consequently, this becomes a future line of research that aims to provide practical strategies and knowledge about intrinsic motivation to be applied in the work of future teachers and coaches in training today.

## Figures and Tables

**Figure 1 behavsci-13-00589-f001:**
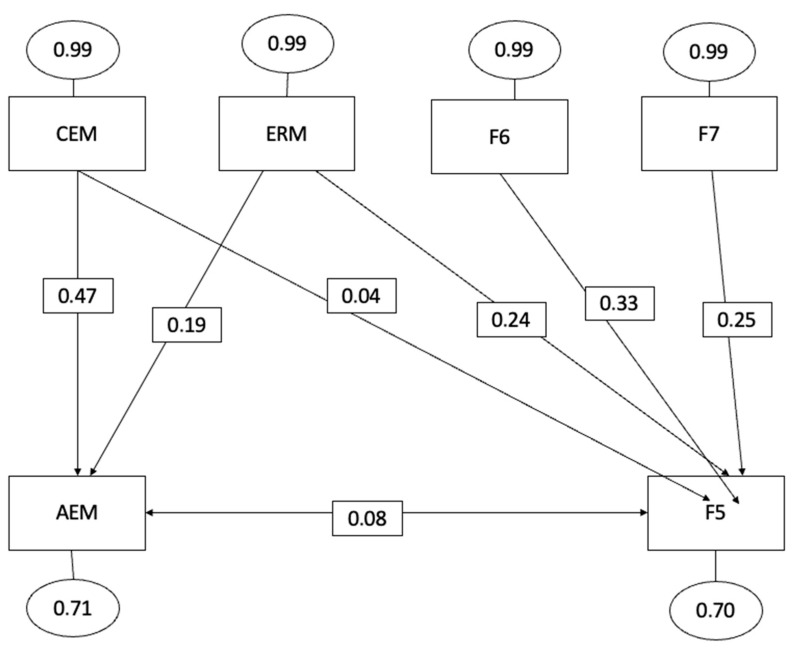
Factor 5 of equational structural modelling with emotional intelligence (Factor 5 intrinsic motivation to experience stimulation).

**Figure 2 behavsci-13-00589-f002:**
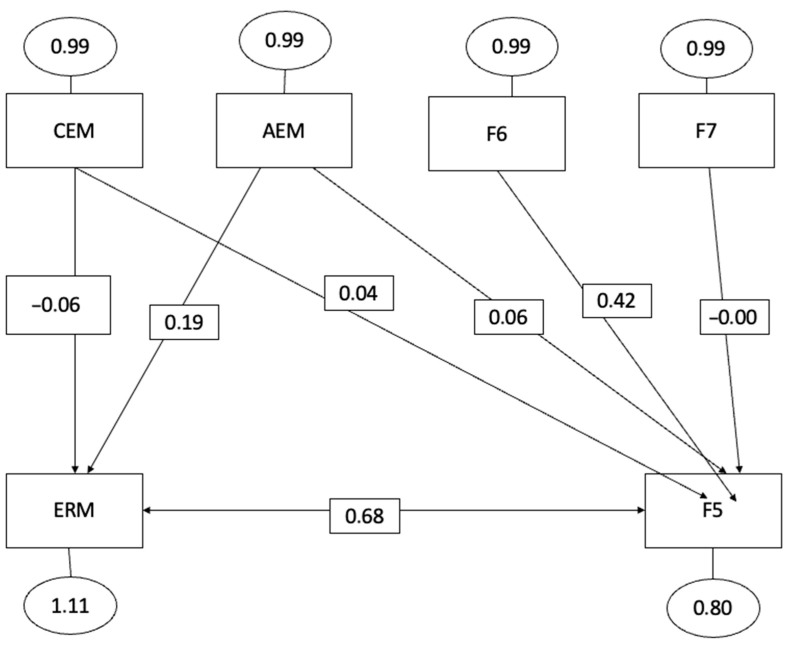
Equational structural modelling factor 6 with emotional intelligence (Factor 6 intrinsic motivation to know something).

**Figure 3 behavsci-13-00589-f003:**
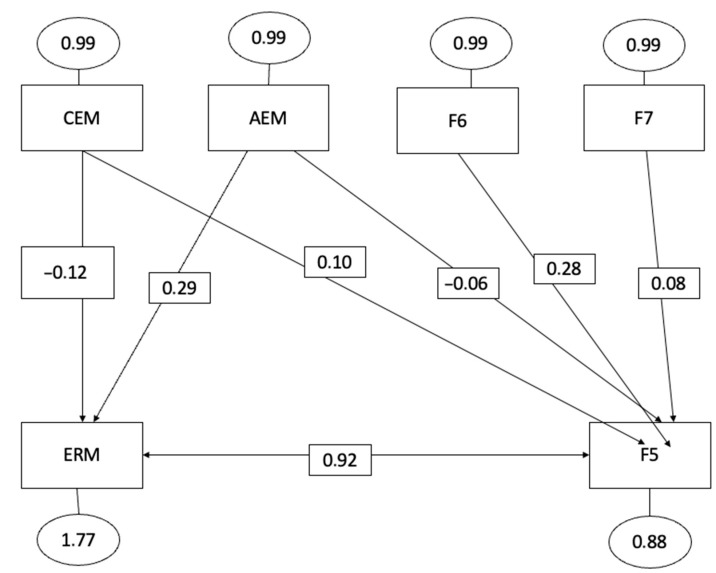
Equational structural modelling factor 7 with emotional intelligence (Factor 7 intrinsic motivation towards knowledge).

**Table 1 behavsci-13-00589-t001:** Description of the sample according to age and sex.

	Females	Males	Total	Under 25 Years	Over 25 Years
First course	18 (39.1)	68 (58.1)	86	86 (97.8%)	2 (2.2%)
Second course	16 (34.8)	23 (19.7)	39	38 (97.4)	1 (2.6%)
Third year	6 (13%)	14 (12%)	20	20 (100%)	0
Total	40	105	145		
Master’s degree	6 (13%)	12 (10.3)	18	4 (22.3%)	17 (77.7%)
Total	46	117	163		

**Table 2 behavsci-13-00589-t002:** Preliminary analysis.

	1	2	3	4	5	6
1. FACTOR 5		0.195 *	0.322 **	0.007	0.115	0.049
2. FACTOR 6			0.430 *	0.230 **	0.122	0.149
3. FACTOR 7				0.316 **	0.128	0.141
4. AE					0.127	0.263 **
5. CE						0.508 **
6. RE						

Note. * *p* < 0.05; ** *p* < 0.01.

## Data Availability

The data presented in this study are available on request from the corresponding author.

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
