# Peer review of "Intrinsic Motivation: Knowledge, Achievement, and Experimentation in Sports Science Students—Relations with Emotional Intelligence"

_behavsci, 2023, doi:10.3390/bs13070589_

Round 1

Reviewer 1 Report

Thank you for your paper. This is an interesting of topic likely of interest to practitioners. There has been a lot of information published on motivation and its role and impact. Additional exploration of how the information in your paper complements work already done would strengthen your paper. As you shared, the sample size is small. Because of the small number of participants, applicability of the findings to a larger population cannot be concluded. Changing the language in the paper to reflect this is very important. You mentioned that should you continue the study, you would instead choose professionals. Additional exploration of why you chose students instead of professionals would further strengthen your paper. 

Please review the paper for sentence structure and continuity. There are several sections where sentence fragments are included which makes the paper challenging to read and understand. 

Author Response

REVIEWER 1

Thank you for your role. This is an interesting topic that is likely to be of interest to professionals. Much information has been published about motivation and its role and impact.

  • Dear reviewer, thank you very much for your contributions which substantially help to improve this work. We appreciate the advice and time spent on this work. All changes are indicated in green.
  1. Further exploration of how the information in your paper complements the work already done would strengthen your paper.
  • It has been added in the introduction that the fact that there is a great lack of research addressing the intersection of intrinsic motivation and emotional intelligence limits our understanding of the psychological processes involved in sport and how emotional intelligence influences these dynamics. By filling this gap, we aim to build a better knowledge base, enabling the development of more informed and effective approaches and strategies to improve and motivate athletes (lines 296-301)
  1. As he shared, the sample size is small. Due to the small number of participants, the applicability of the findings to a larger population cannot be concluded. Changing the language in the document to reflect this is very important.
  • This information has been added in the discussion, indicating that we cannot fail to emphasize that our sample has a very limited number of participants, which does not allow us to generalize these results (lines 699-700). This fact is also referred to in the section on the limitations of the study.
  1. He mentioned that if he continued the study, he would choose professionals. Further exploration of why he chose students over professionals would further strengthen his work.
  • In the section dedicated to the description of the participants, more descriptive information about the sample has been added, also indicating that the main reason for choosing a sample formed by students was to know what the starting situation is, and the type of knowledge that future professionals possess in their initial training. In this sense, we must emphasize that within the Spanish university system, the studies leading to the teaching of physical education are divided into three main areas: the first refers to the official degree in physical education and physical education sciences. sport. The second of them refers to the specific mention in physical education within the studies of the official degree in primary education. And the third of them refers to postgraduate or master's studies, where there is both the official master's degree in physical activity and sports sciences, and the master's degree in teacher training with a mention in physical education. In this case, this research takes as a sample the three main areas of university education in terms of the teaching of physical education, with a distribution of the sample as follows: 76.7% (n = 147) studied degree, either the official degree in Physics, Activity and Sports Sciences, or the official title of Primary Education with mention in Physical Education. While 23.3% (n=18) were enrolled in related postgraduate studies, in this case, the official master's degree in physical activity and sports sciences, and the master's degree in teacher training mention physical education (lines 407-428).

Reviewer 2 Report

This is a very good study, in a field of great interest. The paper is well written and the references seem update.

1. In the abstract - Although the abstract provides a general understanding of the focus and results of the study, it could be improved to provide a more comprehensive and informative summary. Therefore, I have identified the following issues:

- Lack of specific research objectives: The abstract does not clearly indicate the specific objectives or research questions addressed in the study.

- The abstract does not provide additional details about the sample, such as age range, gender distribution, or other relevant characteristics of the participants.

- The abstract does not mention the specific methods or measures used to assess intrinsic motivation and emotional intelligence.

- The abstract does not detail the practical implication or significance of the results.

2.    In the „Introduction” chapter:

- I recommend a clearer and better-structured organization of the introductory text to facilitate understanding and following the presented ideas. For example, the mention of self-determination theory and sports psychology without providing clear explanations of how they are interconnected or how self-determination theory can be applied in the context of sports psychology. This makes it difficult to follow and understand how these concepts relate to and influence athlete motivation. I suggest, for instance, an initial section that presents self-determination theory and its importance in sports psychology, followed by an exposition of how motivation and types of motivation influence athlete performance. This would facilitate the understanding and tracking of the ideas presented in the text.

- There are some sentences that are formulated in an unclear or confusing manner, which can affect the understanding of the information. An example where the sentence is unclear or confusing is the following fragment: "Furthermore, based on this theoretical contribution, three types of motivation are distinguished, corresponding to intrinsic motivation (IM), extrinsic motivation (EM) and amotivation (AM)." The presentation of the three types of motivation can be considered unclear and confusing. The text lacks explanations and clarifications regarding these types of motivation, which can hinder the reader's understanding. To avoid confusion and ensure better comprehension, it would have been useful to provide clear definitions and specific examples for each mentioned type of motivation. This way, the reader would have a better understanding of the distinctions and importance of these types of motivation in the context of sports psychology. Even for specialists, unclear or confusing information can create difficulties in understanding and lead to misinterpretations. A clear structure and coherent expression are always recommended to effectively convey information in an accessible manner for all readers, regardless of their level of expertise. Therefore, I recommend that for the fundamental concepts you work with in your scientific endeavor, provide precise definitions where necessary and formulate ideas in a concise and clear manner to facilitate communication and ensure that the information is correctly understood.

- Certain concepts and ideas are presented in a general or superficial manner. For example, the text mentions self-determination theory as one of the most popular and accepted theories related to motivation in sports. However, no details are provided on how self-determination theory works, what its key principles and concepts are, and how it applies in the context of sports. Another example is the mention of the relationship between emotional intelligence and intrinsic motivation in sports in the introductory text, stating that there is an indirect and positive influence of emotional intelligence on athletes' intrinsic motivation. However, no details are given on how and why this influence occurs, what mechanisms emotional intelligence uses to support intrinsic motivation in sports. This causes the idea to be presented in a general way and does not provide enough detail to support the claim.

- In the introductory text, you use multiple bibliographic references listed together in a block (e.g., [31-32-33-34-35-36-37-38-39-40-41-42]; [48-49-50-51-52]) without providing sufficient explanations or details about their content. This can be problematic because citations and bibliographic references should support and enrich the arguments presented in the text. To address this issue, I recommend providing more information about the specific studies and research mentioned in the text. This will help readers better understand the arguments presented and have confidence in the validity of the information. Therefore, I suggest considering this suggestion and, whenever possible, accepting it. When multiple citations are made, it is important to provide additional information about each cited source or highlight the reasons why these sources are relevant in the context of the discussion. Instead of simply listing the cited sources, it would be better to emphasize how each study or research supports or complements the presented argument.

- In the introduction, the specific research objectives and research questions of the study are not clearly formulated. This can be a limitation in understanding and evaluating the research because readers will not have a clear vision of what is intended to be analyzed and investigated in the study. To address this issue, it is recommended to clearly state the research objectives and research questions at the beginning of the study. This will provide readers with a clear understanding of the purpose and focus of the research. By clearly defining the objectives and questions, the study's scope and direction will be more apparent, allowing readers to assess the relevance and potential impact of the research.

3.    In section 2.1. Method:

- It is mentioned that the study uses two validated and standardized questionnaires as a data collection technique. However, no information is provided about the nature and content of these questionnaires. The lack of details can affect the reader's understanding of how the data was collected and the relevance of the instruments used in the study. It would be helpful to provide a more detailed description of the questionnaires, including sample questions;

- In the section regarding data analysis, it is mentioned that the analysis was descriptive and that certain statistical techniques were used. However, specific information about these techniques is not provided. It would be helpful to offer more details about the statistical methods used, especially regarding the mention of structural equation modeling (SEM) for which no bibliographic reference is given. No references are provided for other techniques either. To address this concern, it is recommended to provide a brief explanation of the statistical techniques used in the study. For example, if SEM was employed, the text should include an overview of how SEM works, its advantages, and its relevance to the research questions. Additionally, information about the variables included in the models, how they were defined and measured, and the steps taken in the analysis should be provided. This could involve discussing the validity and reliability of the measurement instruments used, and any specific considerations taken into account during the analysis.

4.    In the "Results" section, it is noted that the interpretation of the results is not provided in detail. While correlation coefficients between intrinsic motivation factors and emotional intelligence are presented, a more thorough interpretation of these results is lacking. It would be beneficial to offer clearer explanations and implications of the findings, as well as to establish connections with existing literature in the field.

5.    In the "Discussion" section, it is noted that there is a lack of details regarding how all three proposed hypotheses were addressed. The text states that all three hypotheses were fulfilled but does not provide specific information about how they were tested and what specific results were obtained for each hypothesis. It would be helpful to provide more details about the analysis conducted and the specific results obtained for each hypothesis, in order to support the claims made in the discussion. To address this limitation, it is recommended to provide a comprehensive explanation of how each hypothesis was tested and the specific statistical analyses that were employed. Describe the variables involved in each hypothesis, the methodology used to test them, and the statistical tests performed. Additionally, present the specific results obtained for each hypothesis, including any significant findings and effect sizes if applicable. By offering more detailed information about the analysis and specific results for each hypothesis, the discussion will be more robust and supported. This will allow readers to better understand how the hypotheses were addressed and the extent to which the results align with the initial research expectations.

6.    In the research paper, the limitations of the study are mentioned but not discussed in detail. The text acknowledges that the sample size is a limitation and suggests that future research with larger samples would be ideal. However, the impact of this limitation on the validity and generalizability of the results is not thoroughly discussed. Additionally, participant selection appears to have been conducted through convenience sampling, where students who attended the class on a specific day agreed to participate. This raises concerns about the representativeness of the sample. If participants were not randomly selected or do not represent a broader population, the generalizability of the study's results to the target population or other contexts may be limited. It would be beneficial, therefore, to provide a more detailed discussion of the research limitations and how they could influence the interpretation of the results.

7.   In the conclusions, it is emphasized that there is a relationship between the three dimensions of emotional intelligence and intrinsic motivation among university students in the field of physical activity and sports sciences. The conclusions also state that emotional intelligence is of great importance for intrinsic motivation and support the need for future professionals in the field of physical activity and sports sciences to be trained in emotional preparation, not just physical or cognitive preparation. The issue that seems to be unaddressed is that, given that the students in the sample are future physical education teachers or coaches, it can be argued that the main purpose of the research would be to provide them with strategies and knowledge about intrinsic motivation to be applied in their work with future students or athletes, rather than for their own benefit. Thus, the conclusions of the research could have been interpreted in the sense that emotional intelligence and intrinsic motivation are important aspects to be developed and cultivated in future teachers and coaches, in order to help them understand and support the intrinsic motivation of their students or athletes effectively. This could contribute to increasing the engagement, satisfaction, and performance of students and athletes in physical and sports activities. Unfortunately, based on the findings, it is not clear whether this aspect was pursued, which is why it could be included in the limitations or future research directions, where it is currently not mentioned.

Author Response

REVIEWER 2

This is a very good study, in a field of great interest. The document is well written and the references seem up to date.

  • Dear reviewer, thank you very much for your contributions which substantially help to improve this work. We appreciate the advice and time spent on this work. All changes are indicated in yellow.
  1. In summary: Although the summary gives a general understanding of the approach and results of the study, it could be improved to provide a more complete and informative summary. Therefore, I have identified the following problems:

- Lack of specific research objectives: The abstract does not clearly indicate the specific objectives or research questions addressed in the study.

- The abstract does not provide additional details about the sample, such as age range, gender distribution, or other relevant characteristics of the participants.

- The abstract does not mention the specific methods or measures used to assess intrinsic motivation and emotional intelligence.

- The abstract does not detail the practical implication or meaning of the results.

  • The summary has been rewritten taking into account all their contributions. As for the specific objectives, it has been detailed that as specific objectives the analysis of attention, clarity and emotional regulation and the intrinsic motivation to know something are identified. The exploration of the relationship between attention, clarity and emotional regulation and the intrinsic motivation to achieve something. And investigate the correspondence between attention, clarity and emotional regulation and intrinsic motivation to experience stimuli.
  • Likewise, the sample has been detailed in greater depth, indicating that it is composed of 163 undergraduate and master's students related to the Sciences of Physical Activity and Sport, who study at a Spanish public university, located in the southeast of the Spanish country, specifically in Andalusia, in the city of Almería. The mean age was 20.33 years, and in terms of gender, 70.9% (n=117) were male and 27.9% (n=46) female. While according to the degree, 76.7% (n=147) were undergraduate students, and 23.3% (n=18) were graduate students.
  • Regarding the evaluation measures applied, or instruments, it has been indicated that the Sports Motivation Scale (SMS/EMD) has been used to assess intrinsic motivation. While the TMMS-24 was used to evaluate emotional intelligence.
  • In addition, it is indicated that the meaning of the results is to demonstrate the existence of the  relationship between the three dimensions of emotional intelligence (emotional attention, emotional clarity and emotional regulation) and intrinsic motivation (intrinsic motivation to know something, intrinsic motivation to achieve something, and intrinsic motivation to experience stimulation).  Demonstrating that emotional intelligence is claimed as an aspect of great importance for intrinsic motivation (lines 18-40).
  1. In the "Introduction" chapter:

 2.1 I recommend a clearer and better structured organization of the introductory text to facilitate the understanding and follow-up of the ideas presented. For example, mentioning self-determination theory and sports psychology without providing clear explanations of how they are interconnected or how self-determination theory can be applied in the context of sports psychology. This makes it difficult to follow and understand how these concepts relate to and influence athletes' motivation. I suggest, for example, an initial section introducing the theory of self-determination and its importance in sports psychology, followed by a discussion of how motivation and types of motivation influence athletes' performance. This would facilitate the understanding and follow-up of the ideas presented in the text.

  • The information has been reorganized with respect to the Theory of Self-Determination (SDT), arguing that it offers a theoretical model that corresponds to an explanatory theory of human motivation aimed at responding to what are those human behaviors that the subject performs volitionally, or in a self-determined way. That is, the degree to which people perform their actions at the highest level of reflection and engage in actions with a sense of choice (lines 82-94).
  • In addition, information has been added about its importance in sports psychology, where its main contributions lie in the determination of the degree to which athletes are involved or not in sport, from the consideration of different psychological aspects regulating behavior (lines 82-94).
  • Finally, the definition and types of motivation are exposed from these contributions, where motivation is understood as a multidimensional psychological construct that oscillates between different levels of intensity, and that responds to different purposes and intentions, and that can be mediated by different wills, where three types of motivation are distinguished, corresponding to intrinsic motivation (MI), extrinsic motivation (EM) and amotivation (AM) (lines 95-116).

2.2 There are some sentences that are formulated in an unclear or confusing manner, which can affect the understanding of the information. An example where the sentence is unclear or confusing is the following excerpt: "In addition, based on this theoretical contribution, three types of motivation are distinguished, corresponding to intrinsic motivation (MI), extrinsic motivation (EM) and amotivation (AM)". The presentation of the three types of motivation can be considered unclear and confusing. The text lacks explanations and clarifications regarding this type of motivation, which can make it difficult for the reader to understand. To avoid confusion and ensure better understanding, it would have been useful to provide clear definitions and specific examples for each type of motivation mentioned. In this way, the reader would better understand the distinctions and importance of these types of motivation in the context of sport psychology. Even for specialists, unclear or confusing information can create difficulties of understanding and lead to misinterpretations. A clear structure and consistent expression are always recommended to convey information in an effective and accessible way to all readers, regardless of their level of experience. Therefore, I recommend that for the fundamental concepts you work with in your scientific endeavor, you provide precise definitions where necessary and formulate ideas in a concise and clear manner to facilitate communication and ensure that information is properly understood.

  • Each type of motivation has been defined and exemplified. Indicating in this regard that the absolute lack of motivation corresponds to demotivation (which occurs both intrinsically and extrinsically) and corresponds to the lowest level of self-determination, and that at the behavioral level is identified with the null intention of action- A clear example in sports psychology are those situations with which the athlete does not feel able or safe to face. In this sense, those athletes who are at this level practice sports without having an affective, social, or material objective, and experience negative emotions, such as apathy, depression or the feeling of being incompetent (lines 96-104).
  • On the other hand, extrinsic motivation is one in which the engine or force resides in the rewards obtained by performing the behavior, which causes external satisfaction that comes from prizes or rewards. A clear example of this type of motivation in the field of sport corresponds to those people who play sport for social recognition, or for internal or external pressures (lines 104-108).
  • And, finally, intrinsic motivation is one in which the motor or impulse force resides in the pleasure caused by performing the activity, accompanied by feelings of competence and self-realization. Within it, the three subtypes that are analyzed in this study have been identified, which correspond to the intrinsic motivation towards knowledge (understood as that activity that the athlete practices for the pleasure and satisfaction he experiences while trying to learn). The intrinsic motivation towards execution (in which the athlete practices the activity for pleasure while trying to improve or surpass himself). And the intrinsic motivation towards stimulation (in which the athlete practices the activity to experience sensations associated with their own senses) (lines 108-116).

2.3 Certain concepts and ideas are presented in a general or superficial way. For example, the text mentions the theory of self-determination as one of the most popular and accepted theories related to motivation in sport. However, no details are provided on how self-determination theory works, what its key principles and concepts are, and how it applies in the context of sports.

  • Information regarding the Theory of Self-Determination (SDT), as well as its key principles and concepts, and its influence and relationship in the sports field has been rewritten (lines 82-116).

2.4 Another example is the mention of the relationship between emotional intelligence and intrinsic motivation in sport in the introductory text, stating that there is an indirect and positive influence of emotional intelligence on the intrinsic motivation of athletes. However, no details are given on how and why this influence occurs, what mechanisms emotional intelligence uses to support intrinsic motivation in sport. This makes the idea present in a general way and does not provide enough detail to support the claim.

  • Information regarding the relationship between emotional intelligence and intrinsic motivation in sport has been rewritten, indicating first of all the benefits of emotional intelligence in athletes (lines 175-181). And specifically, in terms of the relationship between the three dimensions of emotional intelligence and intrinsic motivation in sport, where intrinsic motivation is associated with both desirable attitudes and values in sports practice, and with better learning. And as a result, it exhibits a direct correlation with greater perseverance on task and greater overall well-being, spanning from childhood to adolescence. As well as that within sports activities and practices, intrinsic motivation is related to a positive attitude towards sport, involvement, active participation, high levels of physical activity, positive social and interpersonal experiences, greater effort and high levels of enjoyment (lines 182-199).

2.5 In the introductory text, use several bibliographic references listed together in a block (for example, [31-32-33-34-35-36-37-38-39-40-41-42]; [48-49- 50-51-52]) without giving sufficient explanations or details about its content. This can be problematic because citations and bibliographic references must support and enrich the arguments presented in the text. To address this issue, I recommend providing more information about the specific studies and research mentioned in the text. This will help readers better understand the arguments presented and have confidence in the validity of the information. I therefore suggest considering this suggestion and, whenever possible, accepting it. When multiple citations are made, it is important to provide additional information about each source cited or highlight the reasons why these sources are relevant in the context of the discussion. Instead of simply listing the sources cited, it would be better to emphasize how each study or research supports or complements the argument presented.

  • Dear reviewer we have tried to comply with this recommendation whenever possible, such as on lines 175-180.
  • On other occasions, as for example in lines 150-153, it has not been possible, since all the references indicated refer to the same idea.

2.6 In the introduction, the specific research objectives and research questions of the study are not clearly formulated. This can be a limitation to understanding and evaluating the research because readers will not have a clear view of what is intended to be analyzed and investigated in the study. To address this issue, it is recommended to clearly state research objectives and research questions at the beginning of the study. This will provide readers with a clear understanding of the purpose and focus of the research. By clearly defining the objectives and questions, the scope and direction of the study will become more apparent, allowing readers to assess the relevance and potential impact of the research.

  • Both the summary and the beginning of the methodology have indicated the main objective and specific objectives (lines 304-321311 As well as the hypotheses have been rewritten in order to improve the reader's understanding (lines 312-321).
  1. In section 2.1. Method:

3.1 It is mentioned that the study uses two validated and standardized questionnaires as data collection techniques. However, no information is provided on the nature and content of these questionnaires. Lack of detail can affect the reader's understanding of how the data was collected and the relevance of the instruments used in the study. It would be useful to provide a more detailed description of the questionnaires, including sample questions;

  • The name, main characteristics, total number of items, and sample questions examples of each of the two standardized instruments included in these investigations have been added. Indicating that the Meta Mood Scale (TMMS 24) [which corresponds to a self-report measure in which self-perceived emotional intelligence composed of 24 items is measured. Theoretically based on skill models, it is organized into 8 items that evaluate both emotional attention, emotional clarity, and emotional regulation based on statements such as: I pay close attention to my feelings... I am clear about my feelings..., or I have a lot of energy when I feel happy.
  • And the second instrument used is the Sports Motivation Scale (SMS/EMD) which measures motivation through seven factors: FACTOR 1 is related to the regulation identified; FACTOR 2 is related to introjected regulation; FACTOR 3 is related to external regulation; FACTOR 4 is related to demotivation; FACTOR 5 is related to intrinsic motivation to experience stimulation; FACTOR 6 is related to intrinsic achievement motivation; FACTOR 7 is related to intrinsic motivation towards knowledge. And it consists of a total of 28 items, which allows a deeper evaluation of motivation, going beyond measuring only intrinsic motivation, versus extrinsic motivation, based on examples of questions such as: I participate in sport for the enjoyment it gives me... I participate in sport for the prestige of being an athlete... I participate in sport so often I wonder as I am not achieving my goals.... (lines 325-397)

3.2 In the section on data analysis, it is mentioned that the analysis was descriptive and that certain statistical techniques were used. However, no specific information is provided on these techniques. It would be useful to provide more details on the statistical methods used, especially with regard to the mention of the structural equation model (SEM) for which no bibliographical reference is given. References are also not provided for other techniques. To address this concern, it is recommended to provide a brief explanation of the statistical techniques used in the study. For example, if SEM was employed, the text should include an overview of how SEM works, its advantages, and its relevance to research questions.

  • This section has been rewritten indicating the purpose of descriptive statistics, which allows exploratory and descriptive analysis of the data. In this case, the different tests were interpreted from an analytical perspective through the statistical indicators established for it.
  • After that, the reasons for using SEM  are detailed, since they correspond to the purpose of evaluating both multiple and crossed dependency relationships, as well as to identify those factors that explain the relationships between the different variables, assuming that from these models, there are no variables more relevant than others, to finally test the hypotheses raised.
  • And the reasons for applying a non-recursive structural model, because while in recursive models the explanation is ordered asymmetrically in a single direction, in non-recursive models relationships appear that reverse the order of causality, establishing reciprocal relationships. Therefore, non-recursive structural models are more realistic than recursive ones (lines 519-522).

3.3 Additionally, information should be provided on the variables included in the models, how they were defined and measured, and the steps followed in the analysis. This could involve discussing the validity and reliability of the measuring instruments used and any specific considerations taken into account during the analysis.

  • Information related to the fact that after the realization of the descriptive statistics, the hypothetical relationships between the latent variables and those observed were established, in this case, between each of the factors chosen from the sports motivation scale: FACTOR 5 is related to the intrinsic motivation to experience stimulation; FACTOR 6 is related to intrinsic achievement motivation; FACTOR 7 is related to intrinsic motivation towards knowledge. And each of the dimensions of emotional intelligence (emotional attention, emotional clarity and emotional regulation).
  • After that, both the bivariances between the previous factors were calculated. As can be seen in Table 2. Before validating the model, the values of different parameters were determined, as well as the indices of goodness or fit. To support or question the proposed model, we consider different fit indices and their adequacy: TLI above 0.95 (Tucker-Lewis index), SRMR (standardized mean square residue) and RMSEA below 0.08 (mean square error). means of approximation) [92] . All these statistical tests were performed with SPSS version 26 and the statistical analysis programmer R. Finally the model was accepted and the hypotheses were accepted, concluding it (lines 523-542).
  1. In the "Results" section, it is noted that the interpretation of the results is not provided in detail. Although the correlation coefficients between intrinsic motivation factors and emotional intelligence are presented, a more complete interpretation of these results is lacking. It would be beneficial to offer clearer explanations and implications of the findings, as well as to establish connections with the existing literature in the field.
  • From the scores obtained, a greater in-depth analysis of the results has been added, as observed from the scores obtained, the three dimensions of emotional intelligence have a direct and positive relationship with the three types of intrinsic motivation: the intrinsic motivation to know something (F5), the intrinsic motivation to know something (F6),  and intrinsic motivation to experience stimulation (F7). These results are similar to those provided  by other previous research which indicated the existing and reciprocal relationship between emotional intelligence and intrinsic motivation understood as one in which the motor or impulse force resides in the pleasure caused by performing the activity, accompanied by feelings of competence and self-realization.   Therefore, based on these results, we can affirm that emotional intelligence is closely related to intrinsic motivation.
  • These results show that those athletes with higher levels of emotional intelligence also show greater pleasure or satisfaction when learning and exploring, when creating or perfecting an action, and feel more comfortable in actions related mainly to art, aesthetics, fun or enjoyment. That is, those athletes who have higher levels of self-determination, at the same time also have higher levels of emotional intelligence, which also demonstrates the relationship between the theory of self-determination and emotional intelligence. Therefore, we speak of a profile of athlete with a positive attitude towards sports practice, highly involved, active, with high levels of physical activity, who relate to others in a positive way, who use more effort and who also obtain high levels of enjoyment (lines 669-689).
  1. In the "Discussion" section, it is noted that details on how the three proposed scenarios were addressed are lacking. The text states that all three hypotheses were met, but does not provide specific information on how they were tested and what specific results were obtained for each hypothesis. It would be useful to provide more details on the analysis carried out and the specific results obtained for each hypothesis, in order to support the statements made in the discussion. To address this limitation, it is recommended to provide a full explanation of how each hypothesis was tested and the specific statistical analyses that were employed.

Describe the variables involved in each hypothesis, the methodology used to test them, and the statistical tests performed.

In addition, present the specific results obtained for each hypothesis, including significant findings and effect sizes, if applicable. By offering more detailed information about the analysis and specific results for each hypothesis, the discussion will be more robust and sustained. This will allow readers to better understand how the hypotheses were approached and to what extent the results align with initial research expectations.

  • About each of the hypotheses, information has been included on the items and variables that compose it. The type of statistical test performed and the methodology followed (lines 701-746).
  1. 1. In the research paper, the limitations of the study are mentioned but not discussed in detail. The text acknowledges that sample size is a limitation and suggests that future research with larger samples would be ideal. However, the impact of this limitation on the validity and generalizability of the results is not thoroughly discussed. In addition, the selection of participants appears to have been done through convenience sampling, where students who attended the class on a specific day agreed to participate. This raises doubts about the representativeness of the sample. If participants were not randomly selected or do not represent a broader population, generalizability of study results to the target population or other contexts may be limited. It would be beneficial, therefore, to provide a more detailed discussion of the limitations of the research and how they might influence the interpretation of the results.
  • This section has been rewritten indicating that although the sample size is small, the results are representative at the local level in which the study was conducted, because the N of the research universe understood as the total conglomerate from which the information is extracted, was a total of 277 participants. To do this, we add the total number of students enrolled in each of the four courses of the official degree in physical activity and sport sciences. The total number of students enrolled in the mention of physical education of the official title of primary education. The total number of students enrolled in the master's degree in physical activity and sport sciences and, the total number of students enrolled in the specialty of physical education within the official master's degree in teacher training. In addition, we use the margin of error calculator, and specify the total size of the population, the desired confidence level (the standard value used by most researchers is 95%). And the sample size. The score obtained was 3.20%. In this sense, the further away the 50% percentage is, the smaller the margin of error (lines 772-812).
  1. The conclusions highlight that there is a relationship between the three dimensions of emotional intelligence and intrinsic motivation among university students in the field of physical activity and sport sciences. The conclusions also affirm that emotional intelligence has a great importance for intrinsic motivation and support the need for future professionals in the field of physical activity and sport sciences to be trained in emotional preparation, not only physical or cognitive. The issue that seems not to be addressed is that, since the students in the sample are future physical education teachers or coaches, it can be argued that the main objective of the research would be to equip them with strategies and knowledge about intrinsic motivation to be applied in their work with future students or athletes, rather than for their own benefit. Thus, the conclusions of the research could have been interpreted in the sense that emotional intelligence and intrinsic motivation are important aspects to develop and cultivate in future teachers and coaches, in order to help them understand and support the intrinsic motivation of their students or athletes. effectively. This could contribute to increasing the engagement, satisfaction and performance of students and athletes in physical and sports activities. Unfortunately, based on the findings, it is unclear whether this aspect was pursued, so it could be included in future research limitations or directions, where it is not currently mentioned.
  • It has been added to the conclusion as a final contribution that our results highlight the relationship between the three dimensions of emotional intelligence and intrinsic motivation among university students in the field of physical activity and sport sciences. Demonstrating that the treatment of emotional intelligence and intrinsic motivation are important aspects to develop and cultivate in future teachers and coaches, in order to help them understand and support the intrinsic motivation of their students or athletes, which could contribute to increase the commitment, satisfaction and performance of students and athletes in physical and sports activities. This fact, in turn, becomes a future line of research in which the main objective has a much more practical character in terms of providing strategies and knowledge about intrinsic motivation to be applied in their work with future students or athletes, to future teachers and coaches in training today (lines 851-861).

Reviewer 3 Report

Dear Authors,

I read with interest the manuscript received for evaluation. Your work is scientifically well argued, the introduction presents in detail the state of research in the field, the purpose of the research and the three working hypotheses are well formulated. Research on university students identifies the relationship/association between the 3 components of intrinsic motivation and emotional intelligence. I can suggest some ideas for improving the initial version of the manuscript:

1. It would be useful to calculate and present the percentage of students included in the research (163 cases) of the total university students of the faculty.

2. 2.4 (Instruments). You used 2 standardized questionnaires in your research (TMMS-24 to assess emotional intelligence and Sports Motivation Scale/F5-F6-F7 to assess intrinsic motivation). It would be easier for readers to understand the study and the results obtained if you included in an appendix the items used and the related Likert scale (1-5, respectively 1-7).

3. It would be interesting if differences between genders and between the two age categories are obtained.................perhaps in a future study with a larger sample.

4. Table 2. How do you explain the insignificant and negative associations between F5/experience stimulation and AE (emotional attention), respectively CE (emotional clarity)?

5. Lines 201-203:  FACTOR 5 is related to the intrinsic motivation to experience stimulation; FACTOR 6 is related to intrinsic achievement motivation; FACTOR 7 is related to the intrinsic motivation towards knowledge. This classification is no longer respected for lines 270-279, and the same factor (knowledge) is mentioned in points a and b:

(a) For intrinsic motivation to know something (F5) it was positively correlated with emotional attention (= 0.04, p < 0.001), emotional clarity (= 0.04, p < 0.001) and emotional regulation (= 0.24, p < 0.001).

(b) Intrinsic motivation to know something (F6) was positively correlated with emotional attention (=0.06, p < 0.001), emotional clarity (=0.04, p < 0.001) and emotional regulation (=0.68, p < 0.001).

(c) Intrinsic motivation to experience stimulation (F7) was positively correlated with emotional attention (= 0.06, p < 0.001), emotional clarity (= 0.01, p < 0.001) and emotional regulation (= 0.92, p < 0.001).

6. Maybe you could also add the name of the factor for figures/graphs 1-3, not just its number, to avoid possible confusion.

7. Citation style for multiple sources: Lines 37-38: [2-3-4-5-6-7-8-9], line 65 [20-21-22-23-24-25-26], line 68 [31-32-33-34-35-36-37-38-39-40-41-42] etc. I checked with Zotero by entering the journal name and it came up [2-9], [20-26], [31-42]. It is possible that by using other referencing software, your version will also be valid. It is only a technical problem, which does not reduce the value of your study.

Author Response

REVIEWER 3

I read with interest the manuscript received for evaluation. Their work is scientifically well argued, the introduction presents in detail the state of research in the field, the purpose of the research and the three working hypotheses are well formulated. Research in university students identifies the relationship/association between the 3 components of intrinsic motivation and emotional intelligence. I can suggest some ideas for improving the initial version of the manuscript:

  • Dear reviewer, thank you very much for your contributions which substantially help to improve this work. We appreciate the advice and time spent on this work. All changes are indicated in blue.
  1. It would be useful to calculate and present the percentage of students included in the research (163 cases) out of the total number of university students in the faculty.
  • Dear reviewer, following your suggestions, in the limitations section we have included that although it is true that the sample is small, the N of the research universe understood as the total conglomerate from which the information is extracted, was a total of 277 participants. To do this, we add the total number of students enrolled in each of the four courses of the official degree in physical activity and sport sciences. The total number of students enrolled in the mention of physical education of the official title of primary education. The total number of students enrolled in the master's degree in physical activity and sport sciences and, the total number of students enrolled in the specialty of physical education within the official master's degree in teacher training.  In addition, we use the margin of error calculator, and specify the total size of the population, the desired confidence level (the standard value used by most researchers is 95%). And the sample size. The score obtained was 3.20%. In this sense, the further the 50% percentage is, the smaller the margin of error becomes. Therefore, although the sample size is limited, and we cannot generalize the results in the general population, if they correspond to representative results at the local level in which the study was conducted. In this sense, it should be noted that within the Spanish university system, the studies leading to the teaching of physical education are divided into three main areas: the first refers to the official degree in physical education and physical education sciences. sport. The second of them refers to the specific mention in physical education within the studies of the official degree in primary education. And the third of them refers to postgraduate or master's studies, where there is both the official master's degree in physical activity and sports sciences, and the master's degree in teacher training with a mention in physical education. These three large areas are what result in the total of 277 participants as N of the universe (lines 414-428 / line 773-812).
  1. 2.4 (Instruments). You used 2 standardized questionnaires in your research (TMMS-24 to assess emotional intelligence and Sports Motivation Scale/F5-F6-F7 to assess intrinsic motivation). It would be easier for readers to understand the study and the results obtained if it included in an appendix the items used and the related Likert scale (1-5, respectively 1-7).
  • Both questionnaires have been added as an attachment in the same form and appearance in which they were supplied to the sample (lines 1176-1178).
  1. It would be interesting to obtain differences between sexes and between the two age categories..... perhaps in a future study with a larger sample.
  • We have added this suggestion to future lines of research (lines 820-821).
  1. Table 2. How do you explain the negligible and negative associations between F5/experience stimulation and EC (emotional attention), respectively CE (emotional clarity)?
  • We have corrected this error, the script that appeared was a typo and does not mean that such a relationship is negative. It now appears properly and corrected.
  1. Lines 201-203: FACTOR 5 is related to intrinsic motivation to experience stimulation; FACTOR 6 is related to intrinsic achievement motivation; FACTOR 7 is related to intrinsic motivation towards knowledge. This classification is no longer respected for lines 270-279, and the same factor (knowledge) is mentioned in points a and b: (a) For the intrinsic motivation to know something (F5) it was positively correlated with emotional attention (= 0.04, p < 0.001), emotional clarity (= 0.04, p < 0.001) and emotional regulation (= 0.24, p < 0.001). (b) Intrinsic motivation to know something (F6) was positively correlated with emotional attention (=0.06, p < 0.001), emotional clarity (=0.04, p < 0.001) and emotional regulation (=0.68, p < 0.001). (c) Intrinsic motivation to experience stimulation (F7) was positively correlated with emotional attention (= 0.06, p < 0.001), emotional clarity (= 0.01, p < 0.001) and emotional regulation (= 0.92, p < 0.001).
  • This fact has been rewritten and revised so that the factors always appear in the same way and alluding to the same concept, in such a way, now factor 5 corresponds to experiencing stimulation. Factor 6 with the intrinsic motivation to know something, and factor 7 with the intrinsic motivation towards knowledge.
  1. Perhaps you could also add the factor name for figures/graphs 1-3, not just its number, to avoid possible confusion.
  • The name of the factor has been added along with the name of each of the tables of the respective figures/graphs 1-3 (lines 642-643 / 645-646 / 651-652).
  1. Citation style for multiple sources: Lines 37-38: [2-3-4-5-6-7-8-9], line 65 [20-21-22-23-24-25-26], line 68 [31-32-33-34-35-36-37-38-39-40-41-42] etc. I checked with Zotero by entering the name of the magazine and it came up [2-9], [20-26], [31-42]. When using other reference software, its version may also be valid. It's just a technical problem, which doesn't reduce the value of your study.

We have reviewed and corrected the appearance of such references.

Reviewer 4 Report

Topic Intrinsic Motivation: Knowledge, Achievement and Experimentation in Sports Science Students. Relations with Emotional Intelligence is interesting. The presented methodology and research results do not raise any objections. However, the results should be described in more detail.

I propose to transfer the research question (lines 127-130) and research hypotheses (lines 137-146) presented in the introduction to the methodological part.

The course of the research should also be supplemented with the sample selection procedure, i.e. from how large a population and how many universities were the 163 respondents selected from. Since this is not a representative group for students of physical education and sports (for Spain? Portugal? - this is unclearly described), the results do not justify drawing conclusions on their general population. The date of the research should also be indicated.

In their conclusions, the authors admit that they plan to extend the research to a larger group as well as qualitative research. In such a case, I suggest taking into account in the methodology both the selection of a representative sample and factors that may interfere with the study, e.g. the sports level of students or the support of significant people.

Recommendation. The thesis should be supplemented in accordance with the above comments.

Author Response

REVIEWER 4

Theme Intrinsic Motivation: Knowledge, Achievement and Experimentation in Sports Science Students. Relationships with Emotional Intelligence are interesting. The methodology presented and the results of the investigation do not raise any objections.

  • Dear reviewer, thank you very much for your contributions which substantially help to improve this work. We appreciate the advice and time spent on this work. All changes are indicated in pink.

However, the results should be described in more detail.

I propose to transfer to the methodological part the research question (lines 127-130) and the research hypotheses (lines 137-146) presented in the introduction.

  • Following his indications, both the general and specific objectives and the hypotheses now appear in the method section (lines 304-321).

The course of the research should also be complemented by the sample selection procedure, i.e. based on the size of the population and from how many universities the 163 respondents were selected. Since this is not a representative group for students of physical education and sports (for Spain? Portugal? - this is not clearly described), the results do not justify drawing conclusions about its general population.

  • The sample has been described in greater depth, indicating that the N of the research universe understood as the total conglomerate from which the information is extracted, was a total of 277 participants. To do this, we add the total number of students enrolled in each of the four courses of the official degree in physical activity and sport sciences. The total number of students enrolled in the mention of physical education of the official title of primary education. The total number of students enrolled in the master's degree in physical activity and sports sciences AND, the total number of students enrolled in the specialty of physical education within the official master's degree in teacher training (lines 773-812).
  • Indicating in the section of participants that it should be noted that within the Spanish university system, the studies leading to the teaching of physical education are divided into three large areas: the first refers to the official degree in physical education and physical education sciences. sport. The second of them refers to the specific mention in physical education within the studies of the official degree in primary education. And the third of them refers to postgraduate or master's studies, where there is both the official master's degree in physical activity and sports sciences, and the master's degree in teacher training with a mention in physical education. In this case, this research takes as a sample the three main areas of university training in terms of the teaching of physical education, with a distribution of the sample as follows: 76.7% (n = 147) studied degree, either the official degree in Physical Sciences, Activity and Sport, or the official title of Primary Education with mention in Physical Education. While 23.3% (n=18) were enrolled in related postgraduate studies, in this case, the official master's degree in physical activity and sports sciences, and the master's degree in teacher training mention physical education (lines 407-428).

The date of the investigation should also be indicated.

  • It has been indicated that the research was carried out during the 2021/2022 school year (line 429).

In their conclusions, the authors admit that they plan to extend the research to a larger group in addition to qualitative research. In this case, I suggest taking into account in the methodology both the selection of a representative sample and the factors that may interfere with the study, e.g. the sports level of the students or the support of significant others. Recommendation. The thesis should be supplemented according to the above comments.

  • It has been added to the conclusion as a final contribution that our results highlight the relationship between the three dimensions of emotional intelligence and intrinsic motivation among university students in the field of physical activity and sport sciences. Demonstrating that the treatment of emotional intelligence and intrinsic motivation are important aspects to develop and cultivate in future teachers and coaches, in order to help them understand and support the intrinsic motivation of their students or athletes, which could contribute to increase the commitment, satisfaction and performance of students and athletes in physical and sports activities. This fact, in turn, becomes a future line of research in which the main objective has a much more practical character in terms of providing strategies and knowledge about intrinsic motivation to be applied in their work with future students or athletes, to future teachers and coaches in training today (lines 851-861).
  • Thank you very much for your recommendations, we hope we have addressed and clarified all your contributions.

Round 2

Reviewer 1 Report

Thank you for your revisions. I appreciate your efforts to address the sections of concern noted after draft one. Consider moving lines 524-569 in the discussion section up to the section addressing methodology. It seems it would be a better fit there, rather than in the discussion. 

Please review the paper for sentence structure, continuity, spelling, and grammar. There are several sections where sentence fragments are included. Addressing those sections will enhance the readability of the paper.

Author Response

Comments and suggestions for authors

Thank you for your reviews. I appreciate your efforts to address the sections of concern noted after draft one. Consider moving lines 524-569 in the discussion section to the section dealing with methodology. It seems that it would fit better there, rather than in the discussion.

Dear Reviewer, thank you for your comments and for revisiting the manuscript. Their suggestions have been taken into account, and lines 524-569 have been moved to the section dealing with methodology. The changes are highlighted in gray.

Comments on the quality of the English language

Please review the paper for sentence structure, continuity, spelling and grammar. There are several sections where sentence fragments are included. Addressing those sections will improve the readability of the document.

Dear reviewer, as you suggest, the structure, spelling and grammar of the manuscript have been revised. Changes are marked in gray.

Reviewer 2 Report

Dear Authors,

Thank you for your work on the article, I feel the paper improved a lot. However, I suggest you to check the grammatical structure of the work. There are some phrases that do not have the correct logic. Example: Demonstrating that emotional intelligence is claimed as an aspect of great importance for intrinsic motivation.

All the best!

Author Response

Comments and suggestions for authors

Dear authors, Thank you for your work on the article, I feel that the document improved a lot. However, I suggest you check the grammatical structure of the work. There are some phrases that don't have the right logic. Example: Demonstrate that emotional intelligence is claimed as an aspect of great importance for intrinsic motivation. Best wishes!

Dear Reviewer, thank you for your comments and for revisiting the manuscript. His suggestions have been taken into account, and as he suggests, the structure, spelling and grammar of the manuscript have been revised. Changes are marked in gray.

Round 3

Reviewer 1 Report

Thank you for your work on this paper.

I appreciate all of the work you've done to prepare your findings for readers. One final suggestion for change: please review the text in lines 400-402. As it's currently written, it repeats itself and the grammar is incorrect. 

Author Response

I appreciate all the work you have done to prepare your findings for readers. One last suggestion for change: check the text on lines 400-402. As it is currently written, it is repeated and the grammar is incorrect.

  • Dear reviewer, thank you very much for your contributions and the time dedicated to the revision of this manuscript. As suggested we have rewritten lines 400-402 avoiding that the same message appears twice. We want everything to be correct.